Accepted at the ICLR 2024 Workshop on AI4Differential Equations In Science

# ON TRAINING PHYSICS-INFORMED NEURAL NETWORKS FOR OSCILLATING PROBLEMS

**Martin Hofmann-Wellenhof, Alexander Fuchs, Franz Pernkopf**
Christian Doppler Laboratory for Dependable Intelligent Systems in Harsh Environments
SPSC Laboratory, Graz University of Technology, Austria
`{m.hofmann-wellenhof,fuchs,pernkopf}@tugraz.at`

## ABSTRACT

Physics-Informed Neural Networks (PINNs) offer an efficient approach to solving partial differential equations (PDEs). In theory, they can provide the solution to a PDE at an arbitrary point for the computational cost of a single forward pass of a neural network. However, PINNs often pose challenges during training, necessitating complex hyperparameter tuning, particularly for PDEs with oscillating solutions. In this paper, we propose a PINN training scheme for PDEs with oscillating solutions. We analyze the impact of sinusoidal activation functions as model prior and incorporate self-adaptive weights into the training process. Our experiments utilize the double mass-spring-damper system to examine shortcomings in training PINNs. Our results show that strong model priors, such as sinusoidal activation functions, are immensely beneficial and, combined with self-adaptive training, significantly improve performance and convergence of PINNs.

## 1 INTRODUCTION

A multitude of practically relevant systems can be modelled and described via partial differential equations (PDEs), creating the demand for fast and reliable PDE solvers in both research and industry. In PINNs (Raissi et al., 2019), PDEs are used as constraints that are implemented via a problem-specific loss function. Through automatic differentiation (Baydin et al., 2018), partial derivatives can be obtained efficiently and are combined to form the PDE. This PDE constraint is then enforced during optimization and provides strong priors to the model without problem-specific architectures. PINNs can solve equations over a continuous domain and obviate the need for expensive mesh creation.

Physics-informed models have successfully been applied to problems in fluid mechanics, solid mechanics, and optics (Wong et al., 2022). However, they are known to require cumbersome hyperparameter tuning, especially if the solution of the learned PDE contains high-frequency oscillations (Wang et al., 2020). Since oscillating phenomena are ubiquitous, e.g. in multi-body systems, an improved training scheme for these types of problems is of particular interest. In recent years, multiple methods for improving the performance of PINNs have been proposed, ranging from domain decomposition (Jagtap & Karniadakis, 2021), over variational formulation of PINNs (Kharazmi et al., 2019) to transfer learning (Desai et al.). Two methods that should be especially suited to improve the convergence behaviour for oscillating problems are sinusoidal activation functions in the initial network layer (Wong et al., 2022) and self-adaptive PINNs (McClenny & Braga-Neto, 2020).

While prior works have motivated the sinusoidal activations from a learning perspective or a general ability to improve high-frequency model behavior (Wong et al., 2022; Sitzmann et al., 2020), recent theoretical advances in this field argue that the neural activation function should optimally also represent a fundamental solution of the PDE (Ranftl, 2022). This result implies an application of the fundamental solution activation in the model's last layer.
With the fundamental solution being challenging to obtain for complex systems, we propose using sinusoidal activation functions repeatedly throughout the model, as it (i) approximates fundamental PDE solutions for oscillating systems as well as (ii) leverages the beneficial learning properties discussed in (Wong et al., 2022).

This paper analyzes the impact of introducing sinusoidal activations throughout the PINN model in combination with self-adaptive PINNs. We analyze its potential for improving accuracy and reducing training time. We also show that for high-frequency problems, employing multiple layers of sine activation functions in PINNs is beneficial, and a single layer might not be enough. This also translates to physics-informed deep neural operator networks trained for multiple initial conditions, where sinusoidal activation functions provide considerable performance gains, as shown in Appendix A.4. Our experiments use the double mass-spring-damper (dMSD) model as a benchmark problem (results for the wave equation can be found in Appendix A.3).

## 2 RELATED WORK

The idea of incorporating prior knowledge as a PDE into a neural network has existed since the 1990s (Dissanayake & Phan-Thien, 1994). However, the method became viable by exploiting automatic differentiation. Raissi et al. (2019) coined the term physics-informed neural networks in 2019 and reignited interest in this method.
The spectral bias of PINNs (Wang et al., 2020), the difficulty in training PINNs, and the subsequent study of potential failure modes during training (Rohrhofer et al., 2022), (Wang et al., 2021a) lead to a multitude of extensions of the original PINN framework. For oscillating problems, two research directions are of particular interest: sinusoidal feature mapping and loss weighting schemes. Multiple works report the performance benefits of using sinusoidal feature mapping. Here, the two main approaches either utilize sinusoidal activation functions (Raissi et al., 2018), (Fang, 2021), (Zobeiry & Humfeld, 2021), (Sitzmann et al., 2020) or employ multi-scale Fourier feature architectures (Wang et al., 2021b). Wong et al. (2022) are the first to systematically study the impact of *sine* activation functions. They advocate employing them in the first layer only and show that using multiple layers with *sine* activation functions performs similarly. However, in (Ranftl, 2022), it is shown that the activation function should represent a fundamental solution of the PDE, which motivates the use of a *sine* activation function in the last layer.

Several methods were proposed to ease the challenge of balancing the data and physics residuals in the loss function. Non-adaptive weighting (Wight & Zhao, 2020) emphasises the importance of fulfilling the initial condition first by scaling the corresponding residual by a predefined factor. Learning rate annealing (Wang et al., 2021a) considers that finding these scaling factors might be challenging and uses gradient statistics during training to balance the different loss terms. Minimax weighting (Liu & Wang, 2021) uses a different balancing approach, updating the network weights via gradient descent and the residual weights via gradient ascent. The weighting scheme in self-adaptive PINNs (McClenny & Braga-Neto, 2020) is closely related to Minimax weighting but differs in scaling not the entire term by one scalar value but every collocation and data point individually.

## 3 METHODOLOGY

**Physics-informed neural networks:** To solve a PDE with a PINN, we assume that the parameters of the PDE are fixed and known. We can then use a neural network to approximate the desired function and model the PDEs by differentiating the network output with respect to the inputs using automatic differentiation.

The total loss, in our use-case, consists of an initial condition residual $\mathcal{L}_{ic}$ and a physics residual $\mathcal{L}_p$, which can be defined as:

$$\mathcal{L}(\boldsymbol{w}, \boldsymbol{\lambda}^{ic}, \boldsymbol{\lambda}^p) = \mathcal{L}_{ic}(\boldsymbol{w}, \boldsymbol{\lambda}^{ic}) + \beta\mathcal{L}_p(\boldsymbol{w}, \boldsymbol{\lambda}^p), \tag{1}$$

with:

$$\mathcal{L}_{ic}(\boldsymbol{w}, \boldsymbol{\lambda}^{ic}) = \frac{1}{N_{ic}} \sum_{i=1}^{N_{ic}} m(\lambda_i^{ic})(y(t_i, \boldsymbol{x}_i) - \hat{y}(t_i, \boldsymbol{x}_i; \boldsymbol{w}))^2, \text{ and}$$

$$\mathcal{L}_p(\boldsymbol{w}, \boldsymbol{\lambda}^p) = \frac{1}{N_p} \sum_{j=1}^{N_p} m(\lambda_j^p) \left( \frac{\partial \hat{y}(t_j, \boldsymbol{x}_j; \boldsymbol{w})}{\partial t_j} + \mathcal{N}[\hat{y}(t_j, \boldsymbol{x}_j; \boldsymbol{w}); \boldsymbol{\gamma}] \right)^2,$$

where $\mathcal{N}[\cdot, \boldsymbol{\gamma}]$ is a nonlinear operator parameterized by $\boldsymbol{\gamma}$ and $y(t_i, \boldsymbol{x}_i)$ are known values of the PDEs solution at $(t_i, \boldsymbol{x}_i)$. Depending on the studied problem, $y(t_i, \boldsymbol{x}_i)$ can be obtained from known initial

conditions, boundary conditions, or measured data. The predicted solution of the PINN with weights $\boldsymbol{w}$ at collocation points $(t_j, \boldsymbol{x}_j)$, sampled over the whole domain, is given by $\hat{y}(t_j, \boldsymbol{x}_j; \boldsymbol{w})$, and $\beta$ is a scaling factor to balance the individual loss terms (Moseley et al., 2020). The self-adaptation weights $\boldsymbol{\lambda}$ are an extension of the original PINN (McClenny & Braga-Neto, 2020), and $m(\lambda) = 1$ for the baseline PINN. The residual $\mathcal{L}_p$ enforces the PDE at the collocation points.

**PINNs with sinusoidal activation functions:** Using a *sine* instead of or in combination with the more common *tanh* as an activation function was shown to improve the trainability of PINNs (Raissi et al., 2018), (Huang et al., 2021). Wong et al. (2022) provide insights into why this might be the case. The *tanh* function is known to saturate for large inputs, which can lead to vanishingly small gradients. Additionally, the more expressive the PINN gets, the more it suffers from an initial bias toward flat output functions due to a near-zero input gradient. Compared to traditional neural networks, this is especially problematic for PINNs because a zero output $\hat{y}(t_j, \boldsymbol{x}_j; \boldsymbol{w})$, minimizes many PDEs. The PINN is trapped in a local minimum where the physics residual is low or even zero and might not be able to minimize the initial condition residual additionally. Another aspect is that sinusoidal patterns often occur in physical phenomena, such as mass-spring-damper systems, wave propagation, or the heat equation. Therefore, using a sinusoidal activation function is valuable for incorporating prior knowledge into the network. Using a *sine* as an activation function in the first layer as proposed in (Wong et al., 2022) leads to the input of the second layer being a weighted sum of sine functions, which can be used to approximate periodic waves. Naturally, the usage of sinusoidal activation functions can be extended to the subsequent layers (Sitzmann et al., 2020). Sinusoidal activation functions should be especially suited for problems with oscillating solutions.

**Self-Adaptive PINNs (SA-PINNs):** SA-PINNs (McClenny & Braga-Neto, 2020) were introduced to ease the challenge of balancing the individual components of the loss function. The residual weights are jointly optimized with $\mathcal{L}$ and do not need to be tuned by hand after the initialization. Since the weights are applied to each training point individually, the network can balance not only the loss terms as a whole but also the individual training samples. The network focuses on training samples with high weights, effectively concentrating on complex regions of $\mathcal{L}$.

To achieve this, the authors introduce the nonnegative and trainable self-adaptation weights $\boldsymbol{\lambda}$ alongside the strictly increasing, differentiable, nonnegative function $m(\lambda)$ defined on $[0, \infty)$. In the loss equations (1), $\lambda_i^{ic}$ and $\lambda_j^p$ are the self-adaptation weights for the initial condition and physics collocation points. The core optimization problem of SA-PINNs is to minimize the loss with respect to the network weights $\boldsymbol{w}$ but maximize it with respect to the self-adaptation weights $\boldsymbol{\lambda}$ (McClenny & Braga-Neto, 2020).

**Training scheme:** In our proposed training scheme, we combine several findings in PINN literature to improve performance, improve convergence speed, and reduce the required amount of hyperparameter tuning for specific problems. Our training scheme for physics-informed models for oscillating problems comprises two main parts: (i) Applications of *sine* activation functions in *all* layers of the model, to more closely approximate the fundamental PDE solution, as well as reducing vanishing gradient effects, and (ii) combining it with self-adaptive weights to overcome local minima during PINN training.

## 4   EXPERIMENTS AND DISCUSSION

We will test the original PINN and the self-adaptive PINN with up to three different activation function configurations in the hidden layers: the usage of *tanh* in all layers, denoted by the token *all-tanh*, *sine* in the first layer and *tanh* in the subsequent layers denoted as *sine-first*, and *sine* in all layers denoted as *all-sine* (detailed configurations in Appendix A). The following set of differential equations describes the double mass-spring-damper system (see also Appendix A, Figure 3):

$$m_1 \ddot{y}_1 = -c_1 y_2 + c_2(y_1 - y_2) - d_1 \dot{y}_2 + d_2(\dot{y}_1 - \dot{y}_2) \tag{2}$$
$$m_2 \ddot{y}_2 = -c_2(y_1 - y_2) - d_2(\dot{y}_1 - \dot{y}_2) \tag{3}$$

In the often-used quarter-car suspension model, $d_1$ and $c_1$ represent the damping and stiffness of the tyre, while $d_2$ and $c_2$ embody the primary suspension of the car. In the following experiments, we

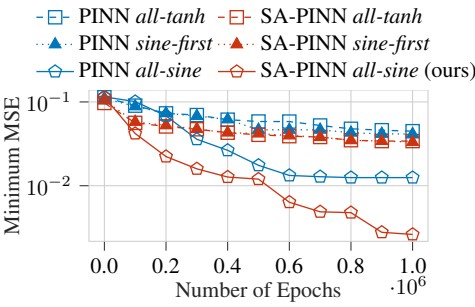
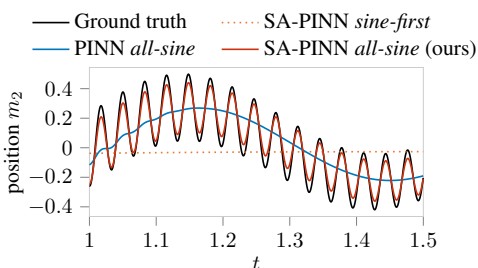

Figure 1: PINN test error for the dMSD model using a single initial condition.

Figure 2: A cutout of the predictions for the dMSD model after $10^6$ epochs for $m_2$.

assume fixed PDE parameters and a fixed initial condition (detailed parameter settings can be found in Appendix A).

Figure 1 shows the minimal obtained test MSE for every 100k of training iterations. It is apparent that using a sinusoidal activation function in all layers is the best-suited approach for this problem. Examining the performance of the *all-tanh* and *sine-first* networks (squares/triangles), we see the results are almost indistinguishable. The introduction of self-adaptive weights (red) improves the performance only marginally.
The *all-sine* networks can distinguish themselves with a considerable performance gap. Notably, the PINN *all-sine* network outperforms two of the three self-adaptive networks. For this problem, the proper activation function configuration outweighs the loss-balancing benefits of the self-adaptive weights. However, the combination of both techniques converges not only the fastest but also to the lowest error overall.

Predictions further away from the initial condition are more challenging for PINNs. Therefore, Figure 2 shows a cutout of the averaged predicted solution after $1e6$ epochs at the end of the temporal domain. One second after the initial condition, networks without *all-sine* activation functions predict a flat output close to zero, here exemplified by SA-PINN *sine-first* prediction, indicating what is likely an attractive local minimum in the loss landscape and illustrating one of the challenges in training PINNs for oscillating problems. Even though the PINN *all-sine* configuration can capture the low-frequency features in the solution after 1 second for both masses, it misses the high-frequency features (see also Appendix A.2, Figure 4). In contrast, SA-PINN *all-sine* can approximate them reasonably well, encouraging the use of our proposed training scheme of combining sinusoidal activation functions in all hidden layers with self-adaptive weights. In Appendix A.4, we demonstrate that the benefits of employing multiple sinusoidal layers extend to deep neural operator networks by solving the dMSD model for various initial conditions without retraining.

## 5    CONCLUSION

In this work, we propose a training scheme for PINNs tailored to oscillating problems. By combining several findings in PINN literature, we are able to improve performance and reduce training time. We show the advantages of applying *sine* activation functions in all layers of the model to more closely approximate the fundamental PDE solution, as well as reducing vanishing gradient effects. In combination with self-adaptive weights, this improves convergence speed during PINN training. Our proposed scheme outperforms all baseline models for the considered double mass-spring-damper model. Here the usage of *sine* activation functions in all, not only the first layer, provides the most benefits.

### ACKNOWLEDGMENTS

The financial support from Siemens Mobility, the Austrian Federal Ministry of Labour and Economy, the National Foundation for Research, Technology and Development, and the Christian Doppler Research Association is gratefully acknowledged.

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

# A  APPENDIX

## A.1  EXPERIMENTAL SETUP

We run each experiment five times with random initialisation for a more objective comparison between the different configurations. We calculate the mean squared error (MSE) over test points sampled evenly over the entire domain and then average the errors over the individual runs. The reference solution is calculated with the help of traditional solvers. We set the simulation parameters to $m_1 = 3e4$, $m_2 = 3e3$, $c_1 = 4e6$, $d_1 = 4e4$, $c_2 = 1e8$, $d_2 = 3e3$ and the initial conditions $y_1 = 0.5$, $y_2 = 1$, $\dot{y}_1 = \dot{y}_2 = 0$. We sample 7,500 fixed collocation points evenly throughout the entire domain ($t \in [0, 1.5]$ seconds) to train PINNs with seven hidden layers with a width of 128. In the output layer, we always use a linear mapping. All networks are optimised with the help of Adam (Kingma & Ba, 2014) (learning rate: $1e-4$), and we set $\beta = 1e-6$ to balance residuals $\mathcal{L}_{ic}$ and $\mathcal{L}_p$.

The tested DeepONet in Section A.4 has 5 hidden layers with a width of 128 in both sub-networks. To match the number of trainable parameters, we increase the hidden layers of the PINN to 9. We define the temporal domain as $t \in [0, 1]$ and set $c_2 = 1e7$. To calculate $\mathcal{L}_p$, we use $5,000$ collocation points distributed evenly over the temporal domain.

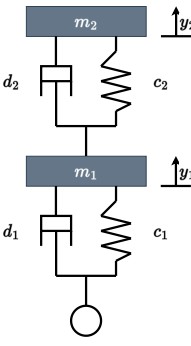

Figure 3: Double mass-spring-damper model. Here, $d_1$, $d_2$, and $c_1$, $c_2$ are the first and secondary damping and stiffness constants.

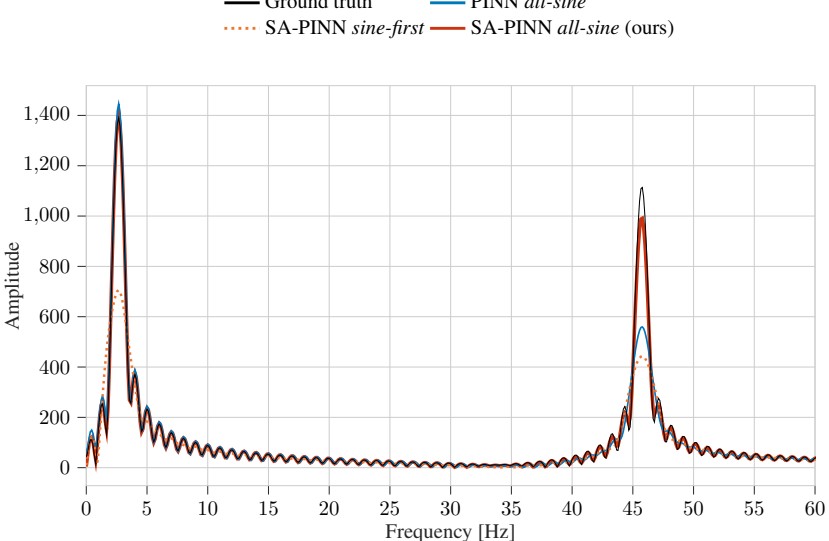

Figure 4: FFT of the true solution and the predictions for $m_2$.

### A.2 DOUBLE MASS-SPRING-DAMPER MODEL FOR A SINGLE INITIAL CONDITION

Figure 4 highlights the performance differences of the individual PINN configurations discussed in Section 4 by showing fast Fourier transformations of the true solution and the predictions for $m_2$. The networks using *all-tanh* or *sine-first* activation functions do not even capture the low frequencies. The PINN *all-sine* network approximates the low frequencies well but misses the strongly oscillating parts of the solution. Only the SA-PINN *all-sine* is able to approximate both ends of the spectrum well, showcasing that employing a single layer with *sine* activation functions does not yield the desired benefits.

### A.3 WAVE EQUATION FOR A SINGLE INITIAL CONDITION

The wave equation occurs in many fields, such as acoustics, electromagnetics, cosmology, and fluid dynamics. We consider the wave equation for a medium with constant density:

$$v^2 \left( \frac{\partial y}{\partial x_1} + \frac{\partial y}{\partial x_2} \right) - \frac{\partial y}{\partial t^2} = 0, \ x_1, x_2 \in [0, 5], \ t \in [0, 10], \tag{4}$$

with the following initial condition:

$$y(0, x_1, x_2) = \exp(-(x_1 - 3)^2 - (x_2 - 3)^2), \tag{5}$$

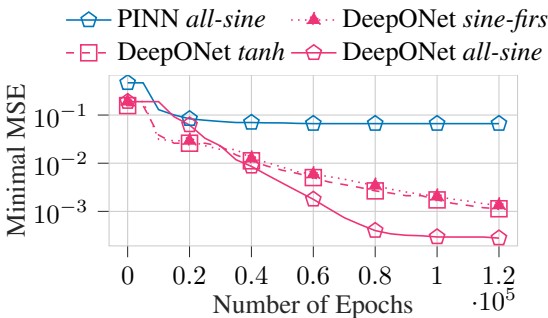

Figure 5: Test error for the double mass-spring-damper model for different initial conditions.

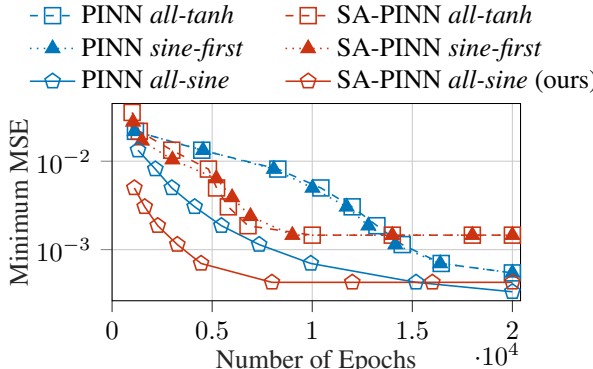

Figure 6: Test error for the wave equation for different initial conditions.

and boundary conditions:

$$\frac{\partial y(t,0,x_2)}{\partial t} = \frac{\partial y(t,x_1,0)}{\partial t} = \frac{\partial y(t,5,x_2)}{\partial t} = \frac{\partial y(t,x_1,5)}{\partial t} = 0, \tag{6}$$

where $v$ is the velocity of the medium, $x_1$, $x_2$ represent the position, and $t$ the time Moseley et al. (2020).

We train a network with four hidden layers of width 256 and apply min-max normalization to the input features. The training dataset incorporates 10,000 samples each for the initial and boundary conditions. To calculate $\mathcal{L}_p$, we use $50,000$ collocation points distributed evenly over the domain and set $\beta = 5e - 4$.

The results in Figure 6 might appear similar to the ones discussed above for the experiments with the double mass-spring-damper model. Once again, the SA-PINNs outperform the baseline PINN in the beginning. However, after about 8,000 epochs, the initially slower converging PINN *all-sine* surpasses it. Therefore, starting with self-adaptive weights and fixing them later in the training process combined with *all-sine* activation functions might be a viable approach for the wave equation. As shown by the results in (McClenny & Braga-Neto, 2020), employing stochastic gradient training is another way of improving the accuracy of SA-PINNs for this particular problem.

### A.4 DOUBLE MSD MODEL FOR MULTIPLE INITIAL CONDITIONS

**Physics-informed deep neural operator networks:** Deep neural operator networks (DeepONets) (Lu et al., 2019) are designed to efficiently learn mappings from a space of functions to another space of functions (operators). Their architecture is based on a universal approximation theorem for operators (Chen & Chen, 1995) and features two sub-networks, a *branch* and a *trunk* network, whose

Table 1: Test error for the double mass-spring-damper model for sampled initial conditions on PINN as well as DeepONet (DON) models, with the classic DON (Wang et al., 2021c) variant using *all-tanh* activations. Here, ID is the MSE of the in-domain data, and OOD is the out-of-domain data.

| MSE | PINN *all-sine* | DON *all-tanh* | DON *sine-first* | DON *all-sine* |
|-----|-----------------|----------------|------------------|----------------|
| ID  | 8.40e-02 | 8.29e-04 | 1.34e-03 | **2.78e-04** |
| OOD | 6.35e-02 | 5.44e-04 | 8.01e-04 | **1.72e-04** |

outputs are combined via a dot product to produce the final result. The branch net encodes the input function. In our experiments with the dMSD system, the input functions will be different initial conditions. The second network, the trunk net, encodes the locations for the output functions. The locations are given by the input time in the dMSD example. Since the outputs of both networks are combined via a dot product, the framework needs to be slightly altered to deal with multiple outputs. In order to obtain outputs for $m_1$ and $m_2$, we split the output neurons of the trunk and branch net in half and compute two dot products as suggested in (Lu et al., 2022). To decrease the input-output pairs needed for training and to produce physically plausible predictions, physics-informed DeepONet were proposed (Wang et al., 2021c). They combine the benefits of DeepONets and PINNs by incorporating an additional residual based on a PDE, as discussed in Section 3, into the loss function of a DeepONet. We will use physics-informed DeepONets in the following experiments but simply denote them as DeepONets. DeepONets are trained to solve the PDE over a predefined input space and only require retraining for out-of-domain samples.

**Experiments and discussion:** Returning to the dMSD model discussed above, we want to assess if the advantages of *all-sine* activation functions persist when using multiple initial conditions. We define the temporal domain as $t \in [0, 1]$ seconds and set $c_2 = 1e7$. We train the networks by sampling 900 initial conditions $y_1, y_2 \in [-1, 1]$ exempt two out-of-domain gaps defined on the intervals $[-0.6, -0.4]$ and $[0.4, 0.6]$. To evaluate the test performance, we randomly sample 100 initial conditions from within the domain and, as a separate metric, 100 initial conditions from the out-of-domain gaps. To calculate $\mathcal{L}_p$, we use $5,000$ collocation points distributed evenly over the temporal domain. The branch net always features *all-tanh* activation functions, whereas we test different configurations of activation functions for the trunk net. Figure 5 illustrates the superior performance the DeepONet architecture enables, compared to PINNs for handling multiple initial conditions. Once again, employing a single layer of *sine* activation functions performs similarly to the original DeepONet *all-tanh*, but DeepONet *all-sine* converges considerably faster than the other configurations. Table 1 shows the MSE for the in-domain (ID) and out-of-domain (OOD) test set after training. In both metrics, DeepONet *all-sine* performs best. Since the MSE depends on the amplitude of the solution, which is influenced by the sampling intervals of the initial conditions, the ID MSE and OOD MSE are not directly comparable. The similar performance of the original *all-tanh* DeepONet and the *sine-first* configuration indicates that also in DeepONets, a single layer with *sine* activations is not enough to profit from the benefits of sinusoidal activation functions.

In future work, we would like to explore if the benefits of self-adaptive weights in PINNs also translate to DeepONets.

