# OpenReview forum: "On training Physics-Informed Neural Networks for Oscillating Problems"
_ICLR.cc/2024/Workshop/AI4DiffEqtnsInSci — AI4DiffEqtnsInSci @ ICLR 2024 Poster_

### Official Review · Reviewer_WHzm · 2024-02-16
**The findings of this work are promising for one case, but I am uncertain about the effectiveness of the method for canonical PDE problems**

**Rating:** 6
**Confidence:** 5

**Review:**

Quality: The quality of this work is satisfactory.

Clarity: Very well-written and easy to follow regarding their aim.

Originality: This framework is somewhat novel, although not entirely unprecedented.

Significance: The findings of this work are promising for one case, but I am uncertain about the effectiveness of your method for canonical PDE problems

---

### Official Review · Reviewer_jz28 · 2024-02-26
**Interesting results but unsure about broader applicability**

**Rating:** 7
**Confidence:** 5

**Review:**

The authors propose using sine activation function for all hidden layers in PINNs. They use SA-PINN as their base model to showcase the proposed usage. The results show the advantage of using sine activation function. However, I have some concerns:

1. The authors should consider comparing with some more activation functions, lets say swish and mish - do they provide accuracy comparable to tanh case?

2. Is it because of the underlying harmonic problem that this fix works? or the authors consider it as widely applicable to other problems?

3. The authors can also combine adaptive activation function while using sine and see if it further improves the learning

4. The paragraphs have different spacing, kindly check that to maintain consistency.

I am happy with the results for acceptance in ICLR workshop because in a way the paper is using the known physical prior in case of a mass damper system to choose the activation function. The choice is intuitive, but is validated empirically and hence would be interesting for this workshop.

---

### Meta-Review · Area_Chair_vgVo · 2024-02-28

**Recommendation:** Accept (Poster)

**Metareview:**

Dear Authors,

Thank you for submitting the draft.

Both reviewers agree that the presented results show some advantages and the draft is clearly written. However, both reviewers do also raise some major points of concern. It is expected that authors will be addressing comments by the reviewers in the final draft.

regards

AC

---

### Decision · Program_Chairs · 2024-02-29

Accept (Poster)